# Safety of tunneled central venous catheters in pediatric hematopoietic stem cell recipients with severe primary immunodeficiency diseases

Illya Martynov[1,2]*, Jessica Klima-Frysch[1], Wolfram Kluwe[1], Christoph Engel[3], Joachim Schoenberger[1]

1 Department of Pediatric Surgery, University Hospital of Freiburg, Freiburg, Germany, 2 Department of Pediatric Surgery, University of Leipzig, Leipzig, Germany, 3 Institute for Medical Informatics, Statistics and Epidemiology, University of Leipzig, Leipzig, Germany

* illya.martynov@medizin.uni-leipzig.de

**Data Availability Statement:** All relevant data are within the manuscript and its Supporting Information files

## Abstract

Tunneled central venous catheters (TCVCs) provide prolonged intravenous access for pediatric patients with severe primary immunodeficiency disease (PID) undergoing hematopoietic stem cell transplantation (HSCT). However, little is known about the epidemiology and clinical significance of TCVC-related morbidity in this particular patient group. We conducted the retrospective analysis of patients with severe PID who received percutaneous landmark-guided TCVC implantation prior to HSCT. We analyzed 92 consecutive TCVC implantations in 69 patients (median [interquartile range] age 3.0 [0–11] years) with severe combined immune deficiency (n = 39, 42.4%), chronic granulomatous disease (n = 17, 18.4%), and other rare PID syndromes (n = 36, 39.2%). The median length of TCVC observation was 144.1 (85.5–194.6) days with a total of 14,040 catheter days at risk (cdr). The overall rate of adverse events during catheter insertion was 17.4% (n = 16) and 25.0% during catheter dwell period (n = 23, catheter risk [CR] per 1000 cdr = 1.64). The most common complication was TCVC-related infection with an overall prevalence of 9.8% (n = 9, CR = 0.64), followed by late dislocation (n = 6, 6.5%, CR = 0.43), early dislocation (n = 4, 4.3%) and catheter dysfunction (n = 4, 4.3%, CR = 0.28). TCVCs are safe in children with severe PID undergoing HSCT with relatively low rates of TCVC-related infection.

## Introduction

Primary immunodeficiency diseases (PID) comprise a wide spectrum of disorders, including more than 350 genetically defined inborn errors of adaptive and innate immunity [1]. Although the clinical presentation of PID is highly variable, many disorders are characterized by recurrent, serious infections, autoimmune dysregulation, and aberrant inflammation if not treated appropriately [2]. The management of PID depends on the type and severity of the underlying defect. Severe forms of PID such as severe combined immune deficiency (SCID) or

**Funding:** The article processing charge was funded by the Baden-Wuerttemberg Ministry of Science, Research and Art and the University of Freiburg in the funding programme Open Access Publishing. The funders had no role in study design, data collection and analysis, decision to publish, or preparation of the manuscript.

**Competing interests:** The authors have no conflicts of interest relevant to this article to disclose

chronic granulomatous disease (CGD) require cellular therapy including hematopoietic stem cell transplantation (HSCT) [3]. The preparation for HSCT and the HSCT procedure itself involves the use of long-term, multilumen, tunneled central venous catheters (TCVCs) for administration of chemotherapy agents during conditioning, infusion of hematopoietic cells, and supportive care management including immunosuppressant agents, antibiotics, analgesics, blood components and parenteral nutrition [4, 5]. Nevertheless, to the profoundly vulnerable immunodepressed HSCT recipients, the TCVCs represents an additional source of morbidity, including procedural adverse events and infective and non-infective complications during catheter dwell period [6–9]. These TCVC-related complications may necessitate additional surgical interventions or culminate in premature catheter removal, both of which coincide with the risks of surgery and general anesthesia while also interrupting or prolonging therapy [7]. Further, it has been shown that infections at the time of HSCT are clearly associated with poorer survival if conditioning is needed [10]. Although the TCVC-related morbidity is highly recognized as a significant problem in pediatric HSCT recipients with severe PID, studies regarding adverse events and complications of TCVCs in this particular patient group are lacking. Therefore, we conducted an observational cohort study which aims to determine the incidence and types of procedural adverse events and dwell period catheter complications in pediatric patients with severe PID undergoing HSCT.

## Patients and methods

### Study population

In this retrospective, single-center observational study, all consecutive HSCT recipients between 0 and 20 years of age with severe PID who underwent implantation of TCVC at the University Medical Center of Freiburg between 01/01/2008 and 31/12/2019 were included. Data on patient and device characteristics, procedural adverse events, and complications during catheter dwell period resulting in revisional surgery or premature removal were retrieved from the Patient Data Management System (PDMS). At the time of data extraction, one of the patients was seeking treatment. All extracted data were fully anonymized before access and subsequently analysis.

### Preoperative evaluation and surgical technique of catheter placement

Preoperative work-up consisted of laboratory studies including coagulation tests. We did not routinely perform ultrasound imaging of the internal jugular vein and carotid artery prior to TCVC implantation. Provided the patient did not receive antibiotic therapy due to the immunodeficiency at the time of the operation, a single-shot prophylactic perioperative antibiotics (e.g. cefuroxim) was administered. In-line filters or anti-infective/microbial lock prophylaxis were not used. Real-time ultrasound guidance for puncture of the vein was not routinely utilized. The implantation was usually performed by surgical residents supervised by board certified pediatric surgeons. Anatomical landmarks and the adapted Seldinger technique were used for catheter cannulation [11]. All catheters were tunneled subcutaneously with the tissue ingrowth cuff positioned at the distal end of the tunnel. Positioning of the catheter tip was confirmed by fluoroscopy.

### Definitions

Procedural adverse events were defined as undesirable conditions during catheter placement, such as complicated guide wire insertion, accidental arterial puncture, multiple unsuccessful venous cannulations, or need for conversion to venae section. Complications during catheter

dwell period requiring catheter tube repositioning, catheter removal or exchange were analyzed. These complications included TCVC-related infection, catheter dysfunction/thrombosis, and early and late catheter dislocation presenting as malposition of the catheter tip. Tip malposition was defined as the position of the tube tip far beyond the right atrial entrance, e.g. within atrium itself (deep malposition), in the superior vena cava (high malposition) or in the ipsi- or contralateral internal jugular or subclavian vein (knocked over malposition). Early dislocation was defined as a catheter tip malposition within the first 7 postoperative days. Late catheter dislocations were classified as events occurring thereafter. Accidental catheter dislocation was also considered to be a dwell period complication. Catheter-related infections were defined as the presence of bacteremia (catheter-related bloodstream infections) and/or local skin inflammation (exit-site infections and/or tunnel infections) originating from the catheter. Catheter dysfunction was defined as an inability to administer the medication or inability to aspirate irrespective of cause (e.g. thrombosis, leakage with extravasation, avulsion of tube fragments).

## Exclusion criteria

Dwell period complications which were treated conservatively or did not require catheter tube repositioning, removal or exchange were excluded from analysis.

## Statistical analyses

Demographic, disease-related, and device-related variables were described using frequencies and percentages for categorical variables. The total number of catheter days at risk (cdr) was calculated as the total number of days from insertion to elective catheter removal or revisional surgery or premature catheter removal. The complication rate per 1000 cdr (CR) was calculated as 1000 times the number of complications, divided by the total number of cdr. Normality of data was evaluated by the Kolmogorov–Smirnov and Shapiro Wilk tests. Results for continuous non parametric data are expressed as median (interquartile range [IQR]). For analyses of continuous variables the Mann-Whitney-U-test was used. All statistical tests were two-tailed and the tests were considered significant with $p < 0.05$.

## Results

### Patient and device characteristics

In total, this study included 92 consecutive catheter positioning procedures in 69 patients with severe PID considered for HSCT. The median age at the time of catheter implantation was 3.0 (0–11) years. The male/female ratio was 2.3 (48 boys and 21 girls). Of the 92 devices inserted, 51 (55.4%) were Groshong and 41 (44.6%) Hickman/Broviac catheters. Of these, TCVC was inserted only once in 49 (71.0%) of the patients, twice in 18 (26.1%) children, three and four times in one patient (1.45%), respectively. The internal jugular vein (n = 89, 96.7%) and the right side (n = 78, 84.8%) were most commonly used. The median time from TCVC implantation to day one of HSCT was 18.5 (13.4–28.6) days. The median length of TCVC observation was 144.1 (85.5–194.6) days and ranged between 5.2 and 588 days for a total of 14,040 cdr. Detailed patients and device characteristics are summarized in Table 1.

### Procedural adverse events and dwell period complications

Overall, 16 (17.4%) adverse events during catheter placement were observed. Among them, complicated guide wire insertion after successful venous cannulation was the most common, with an overall rate of 7.6% (n = 7). Conversion to venous cut-down was necessary in 5 (5.4%)

**Table 1. Clinical and demographic characteristics of patients receiving 92 TCVC implantation.**

| Characteristics | Total |
|---|---|
| **Number of implantations** | n = 92 |
| **Age, y, Median (IQR)** | 3.0 (0–11) |
| **Sex:** | |
| •- Male, No. (%) | 65 (70.7%) |
| •- Female, No. (%) | 27 (29.3%) |
| **Diagnosis, No. (%)** | |
| •- Severe combined immunodeficiency | 39 (42.4%) |
| •- Chronic granulomatous disease | 17 (18.5%) |
| •- Hemophagocytic Lymphohistiocytosis | 9 (9.8%) |
| •- IPEX syndrome | 7 (7.6%) |
| •- LRBA deficiency | 4 (4.3%) |
| •- Hyper IgM syndrome | 4 (4.3%) |
| •- PNP deficiency | 3 (3.3%) |
| •- CTLA4 deficiency | 3 (3.3%) |
| •- Others | 6 (6.5%) |
| **Time to HSCT after TCVC implantation, d, Median (IQR)** | 18.5 (13.4–28.6) |
| **Device type, No. (%)** | |
| •- Groshong | 51 (55.4%) |
| •- Hickman/Broviac | 41 (44.6%) |
| **Tube size (French), No. (%)** | |
| •- ≤ 7.0 | 40 (43.5) |
| •- > 7.0 | 52 (56.5) |
| **Venous access, No. (%)** | |
| •- Internal jugular vein | 89 (96.7) |
| •- Subclavian vein | 2 (2.2) |
| •- External jugular vein | 1 (1.1) |
| **Laterality, No. (%)** | |
| •- Right side | 78 (84.8) |
| •- Left side | 14 (15.82) |
| **Catheter dwell time, d, Median (IQR), catheter days at risk (cdr), Sum** | 144.1 (85.5–194.6) 14,040 |

Abbreviations: IPEX syndrome: Immune dysregulation, polyendocrinopathy, enteropathy, X-linked syndrome; LRBA deficiency: Lipopolysaccharide-responsive beige-like anchor protein deficiency; PNP deficiency: Purine nucleoside phosphorylase deficiency; CTLA4 deficiency: Cytotoxic T-lymphocyte-associated Protein 4 deficiency; Other syndromes included Wiskott–Aldrich syndrome, Leukocyte adhesion deficiency, Activated PI3 Kinase Delta syndrome, Hoyeraal-Hreidarsson syndrome, Zinsser-Cole-Engman syndrome, Cartilage hair hypoplasia.

cases. No catheter insertion procedures were complicated by pneumothorax or hemothorax. No TCVC implantation related mortality occurred. The overall complication rate during catheter dwell period was 25.0% (n = 23, CR of 1.64 per 1000 cdr) with median time to surgical revision (TTR) or premature TCVC removal of 62.9 (13.3–172.1) days. Table 2 summarizes both procedural adverse events and Table 3 dwell period complications. The most frequent complication was TCVC-related infection with an overall prevalence of 9.8% (n = 9, CR = 0.64) occurring in 7 patients. Of these patients, one experienced 3 TCVC-related infections and the remaining six experienced one TCVC-related infection. Overall, 66.6% (n = 6) of

**Table 2. Adverse events during catheter implantation.**

| Adverse events | No. (%) |
|---|---|
| Complicated guide wire insertion after successful venous puncture | 7 (7.6) |
| Inadvertent arterial puncture | 2 (2.2) |
| Multiple unsuccessful venous punctures | 2 (2.2) |
| Conversion to venous cut-down | 5 (5.4) |
| Total number of adverse events | 16/92 (17.4) |

the pathogens causing TCVC-related infections were Gram-positive, while in 11.1% (n = 1) the blood cultures remained sterile in the setting of applied broad-spectrum antibiotics. Fungal infection was observed in two (22.2%) patients (Table 4). There were no significant differences in median times to elective catheter explantation in patients who completed therapy compared with patients experiencing TCVC-related infections necessitating catheter revision or premature explantation (150 [114–201, range: 20–588] vs. 139 [35–209, range: 35–612], p = 0.97) (Fig 1). Further catheter dwell period complications were relatively infrequent, including late dislocation with overall frequency of 6.5% (n = 6, CR = 0.43), followed by early dislocation (n = 4, 4.3%) and catheter dysfunction (n = 4, 4.3%, CR = 0.28).

## Discussion

Our study addressed the poorly investigated epidemiology of TCVC-related morbidity in pediatric HSCT recipients with severe PID. This is, to our knowledge, the first study examining adverse events during catheter implantation and dwell period complications in this particular patient group. We found that both adverse events and dwell period complications are common in our patient cohort, with an overall incidence of 17.4% and 25.0% respectively.

The incidence of adverse events observed during catheter implantation is consistent with the 4.5%–22.0% incidence reported in prior studies of pediatric oncology patients [12–14]. Although ultrasound guidance has been shown to reduce the incidence of failures and inadvertent arterial punctures in pediatric central venous catheterization [15], percutaneous landmark technique was predominantly utilized in our study as this approach is traditional at our

**Table 3. Complications during catheter dwell period.**

| Complications during catheter dwell period | No. (%) |
|---|---|
| Early dislocation, n (%) | 4 (4.3) |
| TTR, d, Median (IQR) | 5.2 (2.6–6.7) |
| Late dislocation, n (%) | 6 (6.5) |
| CR | CR 0.43; cdr 180 |
| TTR, d, Median (IQR) | 22.6 (16.1–62.7) |
| Infection, n (%) | 9 (9.8) |
| CR | CR 0.64; cdr 1733 |
| TTR, d, Median (IQR) | 139 (35–209) |
| Dysfunction, n (%) | 4 (4.3) |
| CR | CR 0.28; cdr 587 |
| TTR, d, Median (IQR) | 98.6 (18.8–322) |
| Total, n (%) | 23/92 (25.0) |
| CR | CR 1.64; cdr 14,040 |
| Time to surgical revision (TTR) (days), Median (IQR) | 62.9 (13.3–172.1) |

**Table 4. Microorganisms isolation in TCVC-related infections (total n = 9).**

| Gram-positive | n = 6 (66.6%) |
|---|---|
| *Enterococus fecalis* | 2 |
| *Staphylococcus epidermidis* | 2 |
| *Staphylococcus haemolyticus* | 1 |
| *Staphylococcus hominis* | 1 |
| **Fungi** | **n = 2 (22.2%)** |
| *Malassezia furfur* | 1 |
| *Candida guilliermondii* | 1 |
| **Sterile blood culture** | **n = 1 (11.1%)** |

institution. In our study, the most frequent procedural adverse event was complicated guide wire insertion after successful venous puncture, which accounted for 7.6%. Although we consistently included this type of adverse events in our analysis, this does not constitute a cannulation failure in the stricter sense. Moreover, it is questionable whether ultrasound guidance

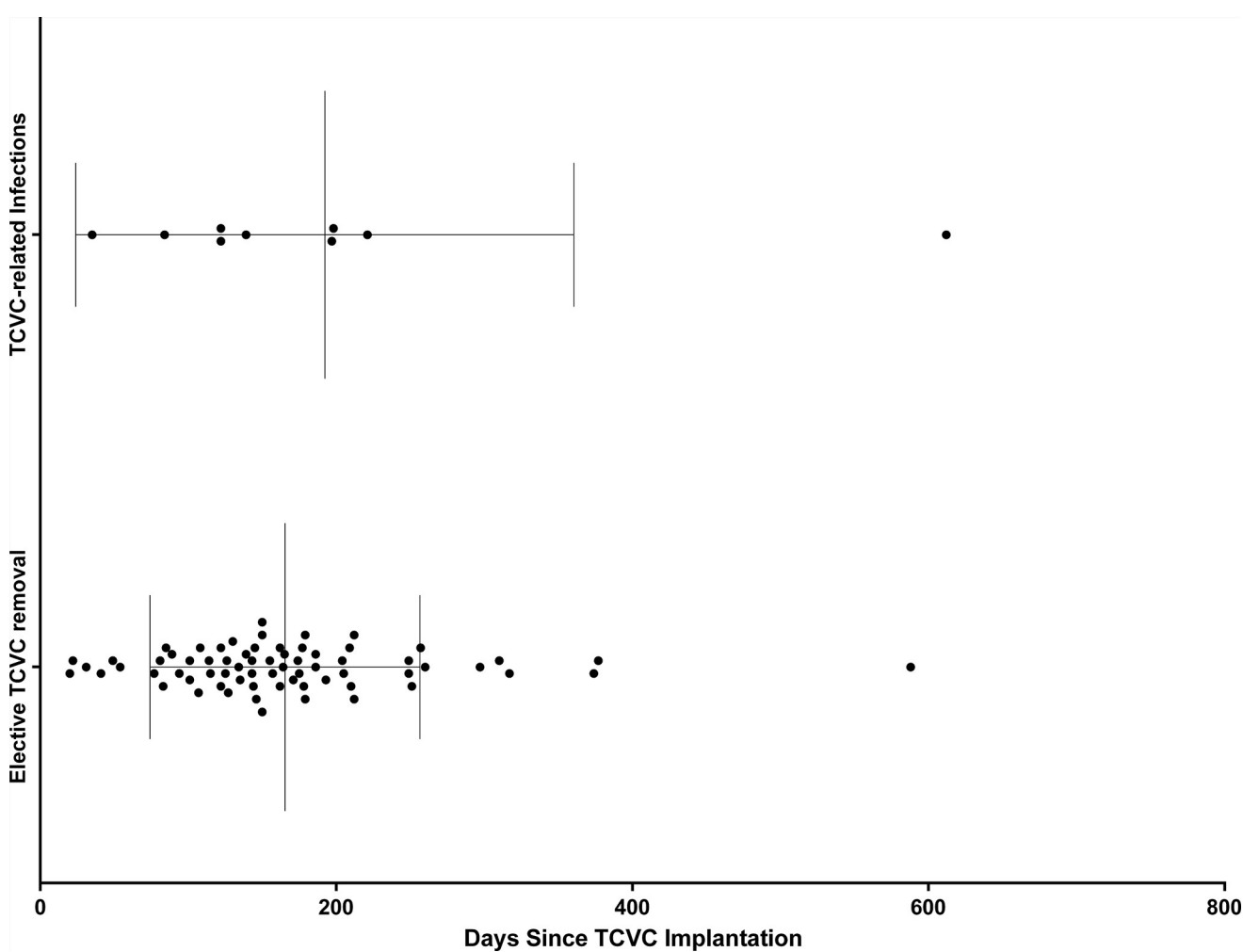

**Fig 1. Elective catheter explantation in patients who completed therapy compared with patients experiencing TCVC-related infections necessitating catheter revision or premature explantation.**

would lower the incidence rate, as the insertion of the guide wire is not ultrasound guided. In contrast, inadvertent arterial and multiple unsuccessful venous punctures represent a typical cannulation failure and the incidence of both could be reduced when using ultrasound [16]. Nevertheless, in our study, the incidence of both complications (inadvertent arterial puncture, n = 2, 2.2%; multiple unsuccessful venous punctures, n = 2, 2.2%) was even lower than that reported in the meta-analysis conducted by de Souza et al., with an overall incidence rate of 5.4% for inadvertent arterial puncture and 5% for unsuccessful venous punctures as reported by others [13, 17]. Finally, in our study population, there were 5 (5.4%) patients requiring conversion from percutaneous catheter insertion to open venous cut-down. Although we classified this mode of tunneled central venous catheter implantation as an adverse event (conversion procedure), this technique is also used as the primary procedure in several centers, especially in small children [18, 19]. In our patients, the conversion cut-down was not associated with further complications. Although the implantation setting, catheter cannulation technique and the experiences of the pediatric surgeons were the same as recently reported by our group investigating TCVC-related morbidity in pediatric oncological patients, the 17.4% incidence of adverse events in the present study was higher compared to 12.8% incidence [20].

Data regarding the incidence of TCVC-related complications during dwell period in pediatric HSCT recipients with PID are lacking. There are only few studies predominantly from the adult literature available investigating TCVC-related morbidity on patients with oncological diseases undergoing HSCT [21–23]. Amongst our patient population, we observed the overall rate of 25.0% (n = 23, CR = 1.64) complications during catheter dwell period. Where comparable, this complications rate is in concordance with the data from a systematic review conducted by Ullman et al. including children with oncological diseases without HSCT and showing that 25% of central vein access devices failed before completion of therapy (CR = 1.97 per 1000 catheter days) [7].

In the current study, the most frequent dwell period complications were TCVC-related infections with the overall incidence rate of 9.8% (n = 9, CR = 0.64) leading to premature catheter removal after median time of 139 catheter in situ days. Even if not directly comparable with the presented data, according to the literature regarding pediatric oncological patients without HSCT, the catheter-associated infections represent the most common complications ranging from 14% to 58% with corresponding CR of 0.57–2.8 episodes per 1000 CVC days at risk [8, 24–27]. The relatively low rates of TCVC-related infection observed in our patients with severe PID undergoing HSCT were highly unexpected due to the fact that these children are per se at very high risk for infections [28]. However, it should be noted that, in this study, we investigated only TCVC-related infections leading to catheter removal, neglecting conservatively treated TCVC-related infection and non-TCVC-related infections. Further non-infective catheter-related complications were very low ranging from 4.3% to 6.5%. These low rates of non-infective complications are in concordance with other studies [29, 30].

Our study has several limitations, including its retrospective nature, single-center setup, and relatively small sample size. The implantation technique and management of TCVC during HSCT are based on local protocols, which may differ from those applied in other centers. Thus, our data on TCVC-related morbidity should be interpreted with caution, and prospective studies are needed. However, while we acknowledge these limitations, our study has also strengths, including patient cohort with very rare diseases.

## Conclusion

This is the first study to date characterizing TCVC-related morbidity in pediatric HSCT recipients with severe PID. We demonstrated that the implantation of TCVC is associated with

17.4% of adverse events and that dwell period complications occur in 25.0% of patients. Unexpected, the incidence of TCVC-related infections was relatively low. The findings of our study provide valuable information on the epidemiology of tunneled central venous catheters indicating their safety in this particular patient group.

## Supporting information

**S1 Dataset.**
(XLSX)

## Author Contributions

**Conceptualization:** Illya Martynov, Joachim Schoenberger.

**Data curation:** Illya Martynov, Wolfram Kluwe, Joachim Schoenberger.

**Formal analysis:** Christoph Engel, Joachim Schoenberger.

**Investigation:** Illya Martynov, Jessica Klima-Frysch, Christoph Engel, Joachim Schoenberger.

**Methodology:** Illya Martynov, Jessica Klima-Frysch, Wolfram Kluwe, Christoph Engel, Joachim Schoenberger.

**Supervision:** Jessica Klima-Frysch.

**Validation:** Jessica Klima-Frysch.

**Visualization:** Jessica Klima-Frysch.

**Writing – original draft:** Illya Martynov, Wolfram Kluwe.

**Writing – review & editing:** Illya Martynov, Jessica Klima-Frysch, Wolfram Kluwe, Joachim Schoenberger.

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
