## [Decision Letter · Decision Letter 0]

16 Mar 2020

PONE-D-20-01609

Safety of Tunneled Central Venous Catheters in Pediatric Hematopoietic Stem Cell Recipients with Severe Primary Immunodeficiency Diseases

PLOS ONE

Dear Dr. Martynov,

Thank you for submitting your manuscript to PLOS ONE. After careful consideration, we feel that it has merit but does not fully meet PLOS ONE’s publication criteria as it currently stands. Therefore, we invite you to submit a revised version of the manuscript that addresses the points raised during the review process.

We would appreciate receiving your revised manuscript by Apr 30 2020 11:59PM. To enhance the reproducibility of your results, we recommend that if applicable you deposit your laboratory protocols in protocols.io, where a protocol can be assigned its own identifier (DOI) such that it can be cited independently in the future. For instructions see: http://journals.plos.org/plosone/s/submission-guidelines#loc-laboratory-protocols

We look forward to receiving your revised manuscript.

Kind regards,

Academic Editor

PLOS ONE

Journal Requirements:

2. In the ethics statement in the manuscript and in the online submission form, please provide additional information about the patient records used in your retrospective study, including:  a) the date range (month and year) during which patients' medical records were accessed; b) the date range (month and year) during which patients whose medical records were selected for this study sought treatment; and c) the source of the medical records analyzed in this work (e.g. hospital, institution or medical center name). If patients provided informed written consent to have data from their medical records used in research, please include this information.

Reviewers' comments:

Reviewer's Responses to Questions

**Comments to the Author**

1. Is the manuscript technically sound, and do the data support the conclusions?

Reviewer #1: Yes

Reviewer #2: Yes

2. Has the statistical analysis been performed appropriately and rigorously? 

Reviewer #1: Yes

Reviewer #2: Yes

3. Have the authors made all data underlying the findings in their manuscript fully available?

Reviewer #1: Yes

Reviewer #2: Yes

4. Is the manuscript presented in an intelligible fashion and written in standard English?

Reviewer #1: Yes

Reviewer #2: Yes

5. Review Comments to the Author

Reviewer #1: The present manuscript is nicely written, detailed analyses of tunneled central venous catheters complications in pediatric patients with severe primary immunodeficiency disease. The major criticism concerns the cannulation technique. Ultrasound was not routinely utilized on cannulations. US is common practice on safe cannulations today. Authors detected more adverse events during cannulations than previously reported. Therefore, adding this technical detain more into discussion would add the scientific value of the present paper.

Study population

All? Consecutive? Recipients?

Approval of the study?

Preoperative evaluation….+ discussion

Ultrasound guidance no routinely utilized? This might be reflected with high adverse events during catheter implantation and therefore results might not reflect present best medical practice. This might be causing the 18% incidence of adverse events detected in the present study compared to other studies? The effect of ultrasound guidance on adverse events has not been discussed in the present manuscript.

Exclusion criteria

How many were excluded?

Reviewer #2: Comments on the manuscript titled ‘Safety of tunneled central venous catheters in pediatric hematopoietic stem cell recipients with severe primary immunodeficiency disease’.

This manuscript examined pediatric patients with hematopoietic stem cell transplantation (HSCT) who require a tunneled central venous catheter (TCVC) in a single center and determined TCVC-related complications and risk factors associated with adverse outcomes.

This study showed adverse events during implantation of TCVC, that has not been examined well so far, as well as dwell period complications, although many studies already reported dwell period complications. Actually, it provided common information or data about complications of TVCV during dwelling period, but as long as the patients with HSCT, this study has something new. It also analyzed correlation carefully between adverse events and patients or device characteristics. Off course, it has limitations such as retrospective nature, too long-term of surveillance and single center setups, the reviewer acknowledges that this is steady investigation as a whole.

<major comments="">

1. Univariate and multivariate analysis were performed even on small number of patients or device characteristics. The reviewer does not think statistically correct.

<minor comments="">

1. Abstract line10: 24.605 � 24,605

  </minor></major>

6. PLOS authors have the option to publish the peer review history of their article (what does this mean?). If published, this will include your full peer review and any attached files.

Reviewer #1: No

Reviewer #2: No

---

## [Author Response · Author response to Decision Letter 0]

22 Apr 2020

Point-by-point response to the comments of the Reviewers

The authors would like to thank the Reviewers for the careful reading of our manuscript and for providing valuable comments and suggestions. The following responses have been prepared to address all your comments in a point-by-point style.

Journal Requirements:

Response 1) 

“1. Please ensure that your manuscript meets PLOS ONE's style requirements, including those for file naming. The PLOS ONE style templates can be found at http://www.plosone.org/attachments/PLOSOne_formatting_sample_main_body.pdf and http://www.plosone.org/attachments/PLOSOne_formatting_sample_title_authors_affiliations.pdf ”

We thank for this comment. We have checked both documents (manuscript file and author´s affiliation file) carefully for style requirements. Both documents have the style required by Plos One Journal.

Response 2) 

“2. In the ethics statement in the manuscript and in the online submission form, please provide additional information about the patient records used in your retrospective study, including: a) the date range (month and year) during which patients' medical records were accessed; b) the date range (month and year) during which patients whose medical records were selected for this study sought treatment; and c) the source of the medical records analyzed in this work (e.g. hospital, institution or medical center name). If patients provided informed written consent to have data from their medical records used in research, please include this information.”

We have amended the methods of the paper as follows: “In this retrospective, single-center observational study, all consecutive HSCT recipients between 0 and 20 years of age with severe PID who underwent implantation of TCVC at the University Medical Center of Freiburg between 01/01/2008 and 31/12/2019 were included. Data on patient and device characteristics, procedural adverse events, and complications during catheter dwell period resulting in revisional surgery or premature removal were retrieved from the Patient Data Management System (PDMS). At the time of data extraction one of patients sought treatment. All extracted data were fully anonymized before access and subsequently analysis.”

Reviewer #1:

Response 1) 

“The present manuscript is nicely written, detailed analyses of tunneled central venous catheters complications in pediatric patients with severe primary immunodeficiency disease. The major criticism concerns the cannulation technique. Ultrasound was not routinely utilized on cannulations. US is common practice on safe cannulations today. Authors detected more adverse events during cannulations than previously reported. Therefore, adding this technical detain more into discussion would add the scientific value of the present paper. Ultrasound guidance no routinely utilized? This might be reflected with high adverse events during catheter implantation and therefore results might not reflect present best medical practice. This might be causing the 18% incidence of adverse events detected in the present study compared to other studies? The effect of ultrasound guidance on adverse events has not been discussed in the present manuscript.”

We thank the reviewer for this comment. We are aware that guided placement of tunneled central venous catheters represents a standard technique nowadays. This recommendation is based on meta-analyses of RCTs comparing real-time ultrasound-guided venipuncture of the internal jugular with an anatomical landmark approach reporting higher first insertion attempt success rates, higher overall success rates, and lower rates of arterial puncture. However, in Europe, and particular in Germany, the ultrasound guided cannulation has not yet been widely adopted in clinical daily practice. In our clinic, we traditionally use landmark technique for the vein cannulation. We agree that the landmark technique represents the limitation of our study and have therefore addressed this point in the discussion section as follows: “The incidence of adverse events observed during catheter implantation is consistent with the 4.5%–22.0% incidence reported in prior studies of pediatric oncology patients [12-14]. Although ultrasound guidance has been shown to reduce the incidence of failures and inadvertent arterial punctures in pediatric central venous catheterization [15], percutaneous landmark technique was predominantly utilized in our study as this approach is traditional at our institution. In our study, the most frequent procedural adverse event was complicated guide wire insertion after successful venous puncture, which accounted for 7.6%. Although we consistently included this type of adverse events in our analysis, this does not constitute a cannulation failure in the stricter sense. Moreover, it is questionable whether ultrasound guidance would lower the incidence rate, as the insertion of the guide wire is not ultrasound guided. In contrast, inadvertent arterial and multiple unsuccessful venous punctures represent a typical cannulation failure and the incidence of both could be reduced when using ultrasound [16]. Nevertheless, in our study, the incidence of both complications (inadvertent arterial puncture, n=2, 2.2%; multiple unsuccessful venous punctures, n=2, 2.2%) was even lower than that reported in the meta-analysis conducted by de Souza et al., with an overall incidence rate of 5.4% for inadvertent arterial puncture and 5% for unsuccessful venous punctures as reported by others [13, 17]. Finally, in our study population, there were 5 (5.4%) patients requiring conversion from percutaneous catheter insertion to open venous cut-down. Although we classified this mode of tunneled central venous catheter implantation as an adverse event (conversion procedure), this technique is also used as the primary procedure in several centers, especially in small children [18, 19]. In our patients, the conversion cut-down was not associated with further complications. Although the implantation setting, catheter cannulation technique and the experiences of the pediatric surgeons were the same as recently reported by our group investigating TCVC-related morbidity in pediatric oncological patients, the 17.4% incidence of adverse events in the present study was higher compared to 12.8% incidence [20].”

Response 2) 

Study population All? Consecutive? Recipients? Approval of the study? Preoperative evaluation….+ discussion. Exclusion criteria How many were excluded?

We appreciate the Reviewer´s comment. We included consecutive pediatric patients with severe PID considered for HSC requiring TCVC implantation as stated in manuscript (page 7, results section). We agree that this information should be mentioned in methods as well. Therefore, we have corrected the text correspondingly: “In this retrospective, single-center observational study, all consecutive HSCT recipients between 0 and 20 years of age with severe PID who underwent implantation of TCVC at the University Medical Center of Freiburg between 01/01/2008 and 31/12/2019 were included. Data on patient and device characteristics, procedural adverse events, and complications during catheter dwell period resulting in revisional surgery or premature removal were retrieved from the Patient Data Management System (PDMS). At the time of data extraction, one of the patients was seeking treatment. All extracted data were fully anonymized before access and subsequently analysis..” There was no need for approval of this study, as all extracted data were fully anonymized before access and analysis. The preoperative evaluation is described in the original version of the manuscript on page 5 (Preoperative evaluation and surgical technique of catheter placement). Regarding the question on exclusion criteria, we agree with the Reviewer that there was no detailed information on patients who were excluded from our study in the initial form of the paper. However, after rewriting the manuscript according to the suggestions of the Reviewer #2 (we now included all consecutive patients between 01/01/2008 and 31/12/2019), we do not have any cases with missing or incomplete data sets or patients with loss of follow up due to catheter removal at other institution. The sentence “Patients with missing or incomplete data or who underwent insertion or catheter removal at other institutions were excluded from analysis” was removed. 

Reviewer #2:

Response 1) 

“This manuscript examined pediatric patients with hematopoietic stem cell transplantation (HSCT) who require a tunneled central venous catheter (TCVC) in a single center and determined TCVC-related complications and risk factors associated with adverse outcomes. This study showed adverse events during implantation of TCVC, that has not been examined well so far, as well as dwell period complications, although many studies already reported dwell period complications. Actually, it provided common information or data about complications of TVCV during dwelling period, but as long as the patients with HSCT, this study has something new. It also analyzed correlation carefully between adverse events and patients or device characteristics. Off course, it has limitations such as retrospective nature, too long-term of surveillance and single center setups, the reviewer acknowledges that this is steady investigation as a whole. 1. Univariate and multivariate analysis were performed even on small number of patients or device characteristics. The reviewer does not think statistically correct. 1. Abstract line10: 24.605 � 24,605”

We would like to thank the reviewer for this comment. We also believe that pediatric hematopoietic stem cell recipients with severe primary immunodeficiency diseases represents a particular patient group in which, currently, no data on the procedural and long-term outcomes of tunneled central venous catheters available. We surely agree with the Reviewer that the retrospective nature, long-term surveillance and the single center setup represent the most important limitations of our study. As “too long-term of surveillance” is the single modifiable parameter, we addressed this suggestion by shortening the observational period to the last 12 years (01/01/2008 – 31/12/2019). We also added two patients with 18 years of age, as well as one with 19 and one with 20 years of age to include all consecutive patients treated in our institution during the last 12 years. The redo analysis showed no significant differences in regard to incidence of adverse events during cannulation (before redo: 18.0% vs. after redo: 17.4%) or dwell period complications (before redo: 24.8% vs. after redo: 25.0%). Thus, without altering the primary outcome parameters of our study, we enhanced the quality of data as all extracted data were complete. We surely agree with the Reviewer that the univariate and multivariate analysis were performed on a small number of patients and device variables affecting the sensitivity (or precision) of the results. The provided OR with corresponding confidence intervals showed high variation. Therefore, to acknowledge the kind suggestions of the reviewer, we have delete both analyses. Finally, 24.605 was corrected to 24,605.

---

## [Decision Letter · Decision Letter 1]

28 Apr 2020

Safety of Tunneled Central Venous Catheters in Pediatric Hematopoietic Stem Cell Recipients with Severe Primary Immunodeficiency Diseases

PONE-D-20-01609R1

Dear Dr. Illya Martynov,

We are pleased to inform you that your manuscript has been judged scientifically suitable for publication and will be formally accepted for publication once it complies with all outstanding technical requirements.

With kind regards,

Academic Editor

PLOS ONE

Additional Editor Comments (optional):

The concerns and issues from the reviewers have been addressed.

Reviewers' comments:

Reviewer's Responses to Questions

**Comments to the Author**

1. If the authors have adequately addressed your comments raised in a previous round of review and you feel that this manuscript is now acceptable for publication, you may indicate that here to bypass the “Comments to the Author” section, enter your conflict of interest statement in the “Confidential to Editor” section, and submit your "Accept" recommendation.

Reviewer #1: All comments have been addressed

Reviewer #2: All comments have been addressed

2. Is the manuscript technically sound, and do the data support the conclusions?

Reviewer #1: Yes

Reviewer #2: Yes

3. Has the statistical analysis been performed appropriately and rigorously? 

Reviewer #1: Yes

Reviewer #2: N/A

4. Have the authors made all data underlying the findings in their manuscript fully available?

Reviewer #1: Yes

Reviewer #2: Yes

5. Is the manuscript presented in an intelligible fashion and written in standard English?

Reviewer #1: Yes

Reviewer #2: Yes

6. Review Comments to the Author

Reviewer #1: Reviewer comments have been addressed adequatelly in the revisioned manuscript. No further comments for the authors

Reviewer #2: The authors have made some correction and improved their manuscript. I have no further questions about this manuscript.

7. PLOS authors have the option to publish the peer review history of their article (what does this mean?). If published, this will include your full peer review and any attached files.

Reviewer #1: Yes: Harri Hakovirta

Reviewer #2: No

---

## [Editor Report · Acceptance letter]

4 May 2020

PONE-D-20-01609R1 

Safety of Tunneled Central Venous Catheters in Pediatric Hematopoietic Stem Cell Recipients with Severe Primary Immunodeficiency Diseases 

Dear Dr. Martynov:

I am pleased to inform you that your manuscript has been deemed suitable for publication in PLOS ONE. Congratulations! Your manuscript is now with our production department. 

With kind regards,

on behalf of

Dr. Robert Jeenchen Chen 

Academic Editor

PLOS ONE